# Prevalence and Associated Factors of HPV Infection in the Oropharyngeal Cavity Among University Students in a Southwest Population in Mexico

**DOI:** 10.3390/diseases14010016

**Published:** 2025-12-31

**Authors:** Joel Jahaziel Díaz-Vallejo, Daniela Córdoba-Colorado, Dulce del Carmen González-Marcial, Ezri Cruz-Pérez, Magda Olivia Pérez-Vásquez, José Locia-Espinoza, Luz Irene Pascual-Mathey

**Affiliations:** Facultad de Química Farmacéutica Biológica, Universidad Veracruzana, Xalapa 91090, Mexico; joediaz@uv.mx (J.J.D.-V.); gs18013822@egresados.uv.mx (D.C.-C.); zs20017550@estudiantes.uv.mx (D.d.C.G.-M.); ezcruz@uv.mx (E.C.-P.); magperez@uv.mx (M.O.P.-V.); jlocia@uv.mx (J.L.-E.)

**Keywords:** human papillomavirus (HPV), oropharyngeal cavity, PCR-RFLP, sexual practices, young adults, transmission

## Abstract

Background: Human papillomavirus (HPV) is the leading cause of sexually transmitted infections (STIs). It is found in extragenital regions, including the oropharyngeal cavity. Its presence in this area is linked to the increased prevalence of oral and pharyngeal cancer cases in young individuals, which is associated with current sexual practices in the young population. Objective, the objective of this study was to estimate the prevalence of HPV infection in the oropharyngeal cavity and identify associated factors within the student community of the Engineering and Chemical Sciences Unit of the University of Veracruz. Methods: an observational, descriptive, and transversal study was conducted. The study included 136 sexually active students aged 18 to 25 without oropharyngeal infection. After obtaining informed consent from all participants, mouthwashes were collected from the oropharyngeal cavity for subsequent detection of viral DNA and HPV genotyping using the PCR-RFLP technique. Risk factors were further assessed through a private questionnaire. For statistical analysis, a bivariate analysis of the main risk factors was performed, and Odds Ratios (OR) and 95% Confidence Intervals (CI) were calculated. Results: The results showed that HPV was detected in 6 participants, resulting in a prevalence of 4.4% (95% CI, 0.92–7.91), with genotypes 11, 52 and 58 identified. Notably, participants with a sexual orientation other than heterosexual had a 7.5-fold higher association with HPV. Conclusions: these findings indicate that low- and high-risk HPV infection in the oropharyngeal cavity is associated with risky sexual behavior in young individuals. Therefore, understanding the specifics of sexual activities is necessary to better comprehend viral transmission and spread among HPV-positive students.

## 1. Introduction

Human papillomavirus (HPV) is the most common sexually transmitted infection (STI) associated with one of the most frequent cancers worldwide, cervical cancer [1]. However, recently, its prevalence has increased in extragenital regions, such as the oropharyngeal cavity, where HPV has been recognized as a risk factor associated with neck and throat illness, specifically oropharyngeal squamous cell carcinoma [2,3]. For instance, it encompasses various neoplasms affecting the mucous membranes surrounding the upper aerodigestive tract, including the oral cavity, oropharynx, hypopharynx, and larynx [3]. In relation to the above, it has been shown that 20% of oral cancer and between 60 and 80% of oropharyngeal cancer are associated with high-risk HPV infection (HPV16/18) [1,4]. However, the prevalence of other high-risk oncogenic factors, such as HPV 52 and 58, has recently been demonstrated [5]. This is extremely important because HPV identification acts as a favorable prognostic factor for oropharyngeal cancer, unlike other factors such as tobacco and alcohol consumption. However, regarding oral cancer, HPV is considered an unfavorable prognostic factor for patients [1,6]. Moreover, it has been shown that tonsillar crypt cells are highly similar to cervical squamocolumnar junction cells, being organized in a discontinuous single-layer epithelium that is more susceptible to carcinogenic transformation [3,4]. This susceptibility may be associated with changes in current sexual practices, particularly deep kissing and oral sex, affecting men more than women, with an incidence range from 2:1 to 4:1, mainly in young individuals under 35 years, which is closely related to the number of sexual partners and contacts [1,3,4,7].

Recent studies have shown that more than 28% of HPV-positive individuals engage in oral sex, including other sexual practices, suggesting that the infection can be transmitted to other anatomical areas, enabling its spread. Remarkably, this is a higher risk of oral HPV transmission [4]. Therefore, it is necessary to know the sexual habits of patients to understand the transmission process [7]. According to the classification of HPV genotypes, group I are carcinogens to humans (like HPV 16, 18, 52, and 58); group 2A are probably carcinogens (HPV 68) and 2B are possibly carcinogens (HPV 26, 53, 66, 67, 70, 73, and 80) to humans, while group 3 are not classified as carcinogens. However, although these genotypes are not directly associated with precancerous lesions, they may be associated with benign conditions in the oral mucosa. Specifically, HPV genotypes 11 and 6 are associated with recurrent airway pathologies, like the laryngeal papillomatosis, with a chronic course [8,9], while HPV 13 is associated with focal epithelial hyperplasia, in juvenile and adults, mainly in the 20–30-year population [3,9,10,11]. In Mexico, more than 50% of young people have their first sexual encounter at an average age of 17.5 years, without using prophylactic measures, which, added to the increase in diverse sexual practices, points to the young population as a susceptible group to acquiring viral infections. Although the infectious process may disappear in approximately two years, some viruses become resistant, while in other cases, reinfections have been observed, mainly of genotypes 16, 18, and 11 [8]. Interestingly, genotypes 6 and 11 have been established as the cause of more than 78 percent of cases of laryngeal papillomatosis, a chronic disease in childhood, adolescence, and adults, coursing without serious consequences or, in extreme cases, surgical intervention is necessary. However, the latter can lead to other complications such as glottic or subglottic stenosis, so it is considered that there are no effective treatments to control or eliminate the disease [8,9]. Therefore, this study aimed to estimate the prevalence of HPV infection in the oropharyngeal cavity, including the associated factors, in a student community at the Engineering and Chemical Sciences Unit of the University of Veracruz. The determination of genotypes in the oropharyngeal cavity and the identification of associated factors to HPV will allow the development of prevention strategies, including healthy sexual practices in the student community.

## 2. Materials and Methods

### 2.1. Participants

A descriptive, observational, and transversal study was conducted in healthy young students. Samples were collected for one week (25 to 29 September) following an “HPV screening campaign” conducted during the month prior to the collection period. The eligibility criteria included healthy adults (18–25 years) enrolled in an undergraduate course at the Engineering and Chemical Sciences Unit of the University of Veracruz from August 2023 to January 2024, who are sexually active. Exclusion criteria were students with a persistent respiratory infection. No incentives were given for participation. Enrolled participants signed an informed consent form (Appendix A), donated a gargle sample, and filled out a questionnaire to evaluate the associated clinical and epidemiological factors (Appendix A. The progress of participants throughout the study is described in the flow diagram shown in Figure 1.

### 2.2. Study Setting

A total of 136 mouthwash samples were obtained by gargling with 10 mL of sterile saline solution (Pisa, Toluca, Mexico) and preserved at 4 °C. This is a suitable sample validated in several studies, representative of oral HPV evaluation [4,12,13]. Briefly, samples were centrifuged (Equipar, CDMX, Mexico) at 6000 rpm for 20 min, and the pellet was resuspended in 400 mL of sterile homogenized saline buffer solution. DNA extraction was performed by the proteinase K method. DNA quantity and purity were analyzed with a UV spectrometer at 260/280 nm (Thermo Fisher, Waltham, MA, USA). A ratio greater than 1.8 was considered adequate for further processing of the samples. The integrity was visualized in a UV light transilluminator (Invitrogen, Carlsbad, CA, USA) through a 1.5% agarose gel electrophoresis (Biorad, Hercules, CA, USA, adding 4 mL of DNA with bromophenol blue dye, obtaining a high integrity in 32.9% of the samples and adequate integrity in the remaining 67.1%.

HPV was identified according to [14], by the amplification of a fragment of the L1 sequence of the viral genome. The samples were subjected to PCR that amplified a 240–250 bp fragment using the oligonucleotides L1C1: 5′CGTAAACGTTTTTTCCCTATTTTTTTT3′ and L1C2: 5′TACCCTAAATACCCTATATTATTG 3′ (IBT-UNAM, CDMX, Mexico) [14]. The reaction mixture consisted of 1X MgCl_2_-free buffer, 1.5 mM MgCl_2_, 200 μM-dNTP, 1.0 μM primers, and 2.0 U Taq Polymerase (Promega, Madison, WI, USA). The thermal cycling program follows the steps: 95 °C for 5 min, 40 cycles of 95 °C for 1 min, 55 °C for 1.5 min, and 70 °C for 2 min, with a final extension step at 72 °C for 5 min (Biorad, Hercules, CA, USA). The ß globin gene was used as an internal control (IBT-UNAM, CDMX, Mexico). Positive HPV samples with LC1/LC2 were typed by RFLP using the restriction enzymes RsaI, PstI, HaeIII, and HinfI, separately, which allows the detection of 9 genotypes (HPV6, HPV11, HPV16, HPV18, HPV31, HPV33, HPV42, HPV52, and HPV58). The final volume was 20 μL (2 μL PCR product, 2 μL restriction buffer, 0.2 μL serum albumin, 0.5 μL Restriction enzyme, and free nucleases water to a final volume of 20 μL) (Promega, Madison, WI, USA). The reaction was incubated in a water bath at 37 °C for 3 h (Equipar, CDMX, Mexico). Inactivation was performed at 4 °C, and samples were visualized on 3% agarose gel stained with ethidium bromide (Biorad, Hercules, CA, USA). The results were verified on a transilluminator (Invitrogen, Carlsbad, CA, USA) by digitizing the gels using Gene Snap software (version 8.0.1.). The restriction map sites used for serotype identification are shown in Table 1 [12,14].

### 2.3. Statistical Analysis

A bivariate analysis of the main risk factors for HPV was performed. Odds ratios (OR) and 95% Confidence Intervals (CI) were obtained to assess if the factors are predictors for HPV prevalence. The level of significance was set as a *p*-value of less than 0.05. All statistical analyses were performed with Statistical Package for Social Sciences version 25.0 (SPSS-IBM) (Armonk, NY, USA: IBM Corp).

### 2.4. Ethical Considerations

The present study was approved by the CONBIOETICA committee of the Dr. Rafael Lucio High Specialty Center (registration number 30-CEI-001-20170221, Appendix A). All procedures were based on the Helsinki Declaration, as well as the Mexican Official Standard NOM-012-SSA3-2012 [15].

## 3. Results

An observational, descriptive, and transversal study was conducted, including 136 sexually active students aged 18 to 25 without oropharyngeal infection. Participants enrolled in the study signed the informed consent form, and mouthwashes were obtained from the oropharyngeal cavity for subsequent detection of viral DNA and HPV genotyping using the PCR-RFLP technique.

### 3.1. Associated Risk Factors to HPV Infection

One hundred thirty-six students participated in the study, with a mean age of 20.64 ± 1.81 years. Of the total population, 108 reported being heterosexual (79.4%, 56 women and 52 men), eight reported being homosexual (5.9%, 2 women and 6 men), 15 were bisexual (11%, 13 women and 2 men), and 5 students (two women and three men) reported another sexual orientation. The age of first sexual encounter ranged from 11 to 22 years, with an average of 17.41 ± 1.96 years. The average number of sexual partners was 3.85 ± 4.77, with a maximum of 32 sexual partners throughout their lives. The women reported an average of 3.75 sexual partners, while men reported 4.25. Regarding sexually transmitted infections, one student reported a previous co-infection of HPV and syphilis, while five participants reported only HPV infection. Of these, four participants reported genital warts. Regarding sexual habits, 74% of women and 49% of men report the absence of prophylactic care during sexual intercourse (condom use). All students have engaged in intimate kissing (kissing with tongue), with an average of 5 intercourses throughout their sexual life. In addition, 67 participants (49.3%) are smokers (34 men and 33 women), 127 participants (93.4%) drink alcohol (58 men and 69 women), and 27 participants (19.9%) use drugs, 14 women and 13 men (Table 2). The only significant risk factor for HPV infection was a sexual orientation other than heterosexual (Table 3).

### 3.2. Amplification and Typification of HPV Using Consensus Primers and Restriction Enzymes

DNA samples extracted from gargle specimens were amplified using universal oligonucleotides L1C1 and L1C2-2, resulting in six positive cases (4.4%) from all samples. HPV serotypes 11 (1.47%), 52 (0.7%), and 58 (2.20%) were identified by RFLP (Table 4, Figure 2).

## 4. Discussion

HPV is responsible for 25% of head and neck cancer cases. In addition, each year, more than 93,000 new patients are diagnosed with Oral and oropharyngeal squamous cell papillomas [16], which are benign tumors associated with more than 25 human papillomavirus (HPV) genotypes. These lesions are typically identified in young adults, although transmission often occurs at an earlier age [17]. The genotypes most frequently associated with oncogenic transformations are the high-risk genotypes (17, 19); however, the genotypes found in normal oral mucous membranes are 6 and 11, which are associated with latent infection. Moreover, genotypes 11 and 13 are associated with squamous cell carcinoma and condylomas acuminata, while genotype 61 is related to oral warts [9]. In the present study, a prevalence of 4.4% of VPH was identified in the oropharyngeal cavity, consistent with the findings of Kreimer et al. [18] who reported a 4% prevalence among young individuals in Brazil, Mexico, and the United States, and by [8], who reported a prevalence of 4.25% in the oral cavity in the Mexican student population. Furthermore, in our population, the average age of first sexual intercourse was 17.4 years; 49% of men and 74% of women reported the absence of prophylactic methods, which agrees with the previously reported, where 50% of students started their sexual life at the age of 17.5 years, which is consistent with the findings of this study [8].

Although it has been reported that the main genotypes associated with the oropharyngeal cavity are HPV16 and 18, in the present study we identified the genotypes 52, 58, and 11, of which two genotypes were considered as high-risk (52 and 58), and one genotype as low-middle risk-factor (HPV11) in young students, which agrees with previously reported by [5,8]. This is particularly important because it has recently been reported that genotypes 52 and 58 may be involved in oncogenesis and transformation in the oral cavity, meaning that their prevalence may influence the development of oral cancerous lesions. Therefore, it is necessary to identify these genotypes in the population, in addition to genotypes 16 and 18 [5].

Moreover, several studies have shown a strong association between early sexual activity, the number of sexual partners, and the lack of contraceptive methods (condoms) with the risk for HPV infection [19]. Among sexual habits, oral sex has been strongly associated as a risk factor for HPV infection [20]. Moreover, “deep kissing” is a risk factor in studies involving young adults aged 18 to 30 [10,21]. Habits such as smoking and alcohol consumption have also been associated with the prevalence of oropharyngeal HPV [22]. Smoking cigarettes had a prevalence rate of up to 11.5% associated with HPV [23]. However, no association was found between smoking and the prevalence obtained in the present study [10].

The primary limitation of this study was the sample size, which restricted the ability to capture greater heterogeneity and to identify stronger associations between risk factors and HPV infection. PCR-RFLP is a suitable technique for detecting HPV genotypes compared to commercial and genotyping kits. This study employed a transversal design, so it did not address the progression of infection. Longitudinal studies examining viral incidence are necessary to provide a more comprehensive understanding of HPV infection dynamics. In summary, the presence of low and high-risk genotypes was detected in young, healthy students. The implementation of public policies that facilitate timely diagnosis, including consideration of sexual practices, is essential to improve access to prevention of oral lesions based on anatomical HPV serotypes.

## 5. Conclusions

Human papillomavirus was detected in the oropharyngeal cavity using the polymerase chain reaction technique, with a prevalence of 4.4% among students, highlighting the importance of STI prevention campaigns among the population. Using the polymorphic restriction fragment length technique, we identified three genotypes (HPV11, HPV52, HPV58). Their detection is important for preventing diseases such as oral cancer, oral focal epithelial hyperplasia, squamous cell papilloma, and condylomas. The interaction of risk factors causes a 7.5-fold increase in HPV infection in the oropharyngeal cavity when sexual orientation is other than heterosexual, representing a risk factor in the study population.

## Figures and Tables

**Figure 1 diseases-14-00016-f001:**
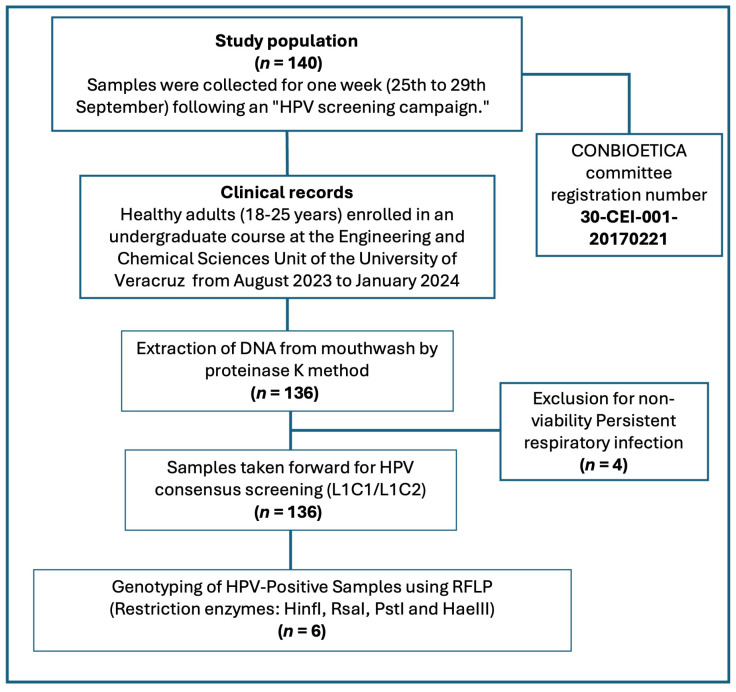
Flow diagram of the study. Data from 136 students were analyzed for HPV by PCR-RFLP. The diagram shows the progress of participants throughout the study.

**Figure 2 diseases-14-00016-f002:**
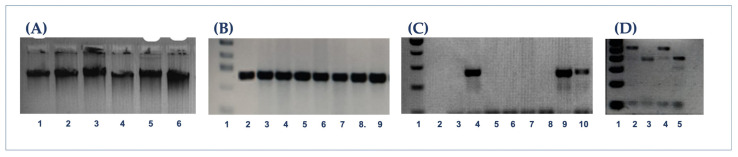
(**A**) 1.5% agarose gel showing the integrity of 6 DNA samples extracted using the proteinase K method. (**B**) 3% agarose gel showing the internal control of β-globin; lane 1, 100 bp marker; lanes 2–9, representative samples. (**C**) HPV-positive samples 244–256 bp. Lane 1: 100 bp marker; lane 3, negative control; lane 4, positive control; lanes 5–8: HPV-negative samples; Lanes 9–10: HPV-positive samples, 2% agarose gel. (**D**) Digestion with endonucleases. Lines 2–5: Enzymes Pstl, Rsal, Hinfl, Haelll, respectively; line 1, 50 bp marker, 3% agarose gel.

**Table 1 diseases-14-00016-t001:** Restriction Enzymes Useful for Typing L1-PCR Product and the Lengths of RFLP.

Enzyme	HPV6	HPV11	HPV16	HPV18	HPV 31	HPV 33	HPV42	HVP52	HPV58
RsaI	-	204, 40	-	-	216, 40	140, 62	-	190, 70	195, 65
HinfI	-	147, 97	-	141, 112	-	-	-	-	-
HaeIII	207, 37	207, 37	200, 53	210, 43	122, 91, 43	112, 91, 43	140, 60	190, 50	200, 40
PstI	-	-	-	190, 63	-	-	-	-	-

The lengths of restriction fragments of the sequenced HPV types were obtained by MAPSORT, according to [14].

**Table 2 diseases-14-00016-t002:** General characteristics of the population.

Population	*N* = 136		
Characteristics	Percentage (n, %)	Men	Women
Heterosexual	108, 79.4%	52	56
Homosexual	8, 5.9%	6	2
Bisexual	15, 11%	2	13
Other sexual orientation	5, 3.7%	3	2
Smoking	67, 49.3%	34	33
Drinking alcohol	127, 93.4%	58	69
Drug use	27, 19.9%	13	14

**Table 3 diseases-14-00016-t003:** Factors associated with HPV prevalence in the oropharyngeal cavity.

Characteristics	OR (CI 95%)	*p*
Sexual orientation other than heterosexual	7.5 (1.0–56.1)	0.048 *
Number of sexual partners	0.3 (0.0–3.0)	0.361
Use of condoms during sexual intercourse	3.5 (0.5–24.7)	0.201
Early onset of sexual activity	1.5 (0.1–13.4)	0.697
Alcohol consumption	0.1 (0.0–2.3)	0.172
Tobacco consumption	0.6 (0.0–5.1)	0.700

* Significant (at 95%); OR: Odds ratio; CI: Confidence interval.

**Table 4 diseases-14-00016-t004:** Frequency of genotypes found.

HPV Risk	HPV Genotypes	Frequency	Percentage n = 136	Infections Associated	Reported by
Low-middle	11	2	1.47%	Squamous cell papilloma (SCP) and condylomas Respiratory papillomatosis	[5,8,9,10]
High	58	3	2.2%	Oral dysplasia and cancer	
High	52	1	0.7%	Oral dysplasia and cancer	

HPV: Human papillomavirus; SCP: Squamous cell papilloma.

## Data Availability

The raw data supporting the conclusions of this article will be made available by the authors on request.

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
