# Peer review of "Prevalence and Associated Factors of HPV Infection in the Oropharyngeal Cavity Among University Students in a Southwest Population in Mexico"

_diseases, 2025, doi:10.3390/diseases14010016_

Round 1

Reviewer 1 Report

Comments and Suggestions for Authors

Currently, HPV carriage is common in one in 10 people (at some point in their life). The high rate of spontaneous elimination of the virus from the host's body, provided the host's immune system is functioning properly, has been proven.

However, in recent years, an increase in the incidence of HPV transmission through non-classical routes has been observed. Furthermore, an increase in the incidence of head and neck cancer has been observed among young patients (under 40 years of age), whose association with this virus is no longer in doubt.

Consequently, it is worth noting that the topic of this study is timely and relevant. This article addresses the important issue of the prevalence of human papillomavirus (HPV) infection in the oropharyngeal cavity among university students in Mexico.

However, the study is not without significant shortcomings.

  1. Introduction

1) Some association of high-risk and low-risk HPV types (HPV16/18 (associated with cancer) and HPV6/11/13/61 (associated with benign lesions)) with head and neck diseases is described. Further results only consider low-oncogenic types. I recommend formulating the problem so as to focus on low-risk genotypes, which is consistent with the study results.

2) Also, if the study focuses on age and geographic location, these parameters should be analyzed in the literature. A more detailed explanation is needed. I recommend including epidemiological data and the results of similar studies conducted in Mexico or other Latin American countries over the past 5 years.

  1. Materials and Methods

1) Justify the choice of biological material and its research methods.

  • No description is given of how the quality and quantity of DNA from the samples were verified.
  • Some sources indicate that the use of L1C1/L1C2 primers and the PFLP method is outdated, has low sensitivity, and a high probability of false positive and false negative results.
  • Were the samples from each subject duplicated?
  • What HPV types were determined? A list is required.
  • The use of the presented methodologies requires confirmation.

2) Multiple logistic regression requires a minimum of 10 events per variable, so they chose an inappropriate statistical method. Recommend using bivariate analysis instead of logistic regression, or increasing the sample size to ensure a sufficient number of HPV cases.

  1. Results

1) The sample varied in the parameter "Sexual orientation other than heterosexual." Therefore, statistical calculations in the overall sample are invalid.

2) The quality of Figure 2 is unsatisfactory for analysis.

The figure legend lists types 33, 35, and 39. How does this relate to Table 2?

Author Response

Comments 1: [Currently, HPV carriage is common in one in 10 people (at some point in their life). The high rate of spontaneous elimination of the virus from the host's body, provided the host's immune system is functioning properly, has been proven.

However, in recent years, an increase in the incidence of HPV transmission through non-classical routes has been observed. Furthermore, an increase in the incidence of head and neck cancer has been observed among young patients (under 40 years of age), whose association with this virus is no longer in doubt. Consequently, it is worth noting that the topic of this study is timely and relevant. This article addresses the important issue of the prevalence of human papillomavirus (HPV) infection in the oropharyngeal cavity among university students in Mexico.

However, the study is not without significant shortcomings]

Response 1: [We appreciate your valuable feedback on this work. Your comments and observations have significantly contributed to enhancing the quality of our research. We trust that shortcomings have been addressed, and that the information provided is now clear]

Comments 2: [Introduction. 1) Some association of high-risk and low-risk HPV types (HPV16/18 (associated with cancer) and HPV6/11/13/61 (associated with benign lesions)) with head and neck diseases is described. Further results only consider low-oncogenic types. I recommend formulating the problem so as to focus on low-risk genotypes, which is consistent with the study results.]

Response 2: [Agree. Based on your suggestions, a new analysis of the results was carried out in accordance with the restriction patterns previously reported by Yoshikawa. This allowed us to identify two high-risk variants within the study population, 52 and 58, which were included in the information presented in the introduction section.]

Comments 3: [Introduction. 2) Also, if the study focuses on age and geographic location, these parameters should be analyzed in the literature. A more detailed explanation is needed. I recommend including epidemiological data and the results of similar studies conducted in Mexico or other Latin American countries over the past 5 years.]

Response 3: [Agreed. Thank you for the recommendation. Updated information on the epidemiological context was included, covering the Mexican population.]

Comments 4: [Materials and Methods

1) Justify the choice of biological material and its research methods.

  • No description is given of how the quality and quantity of DNA from the samples were verified.
  • Some sources indicate that the use of L1C1/L1C2 primers and the PFLP method is outdated, has low sensitivity, and a high probability of false positive and false negative results.
  • Were the samples from each subject duplicated?
  • What HPV types were determined? A list is required.
  • The use of the presented methodologies requires confirmation.]

Response 4:

·      No description is given of how the quality and quantity of DNA from the samples were verified.

Response: [Thanks for pointing that out. The quality and quantity of DNA in the samples are described in the methodology section.]

·      Some sources indicate that the use of L1C1/L1C2 primers and the PFLP method is outdated, has low sensitivity, and a high probability of false positive and false negative results.

Response: [Thank you for pointing that out. For this study, we relied on previous references indicating that this method is both appropriate and sensitive for identifying viral DNA and genotyping. The methodology includes references that validate the use of these techniques.]

·      Were the samples from each subject duplicated?

Response: [Although the results were not replicated, we are confident that they are adequate. The reported procedures are validated and standardized, and we have used techniques and procedures previously validated and reported in different studies included in the methodology.]

·      What HPV types were determined? A list is required.

Response: [Thank you for your comment. The list of HPV types identified with restriction enzymes is included, as well as a table showing the fragments generated by these enzymes.]

·      The use of the presented methodologies requires confirmation.

Response: [Thank you very much for the observation. In the methodology section, we have included the references that confirmed the use of the methodologies proposed in the present manuscript.]

Comments 5: 2) Multiple logistic regression requires a minimum of 10 events per variable, so they chose an inappropriate statistical method. Recommend using bivariate analysis instead of logistic regression or increasing the sample size to ensure a sufficient number of HPV cases.

Response 5: [We greatly appreciate this observation. Taking your recommendation into account, the analysis was adjusted to include only the bivariate analysis, instead of the multiple logistic regression, which is already included in the methodology.]

Comments 6: Results

1) The sample varied in the parameter "Sexual orientation other than heterosexual." Therefore, statistical calculations in the overall sample are invalid.

2) The quality of Figure 2 is unsatisfactory for analysis.

3) The figure legend lists types 33, 35, and 39. How does this relate to Table

Response 6: Results

1) The sample varied in the parameter "Sexual orientation other than heterosexual." Therefore, statistical calculations in the overall sample are invalid

Response: [Thank you very much for your comment. We consider sexually active participants those who have an active sex life, which currently includes sexual relations that may occur vaginally, anally, or orally. Therefore, all participants were considered, because one of the objectives of this study was to establish if the use of new sexual practices conditions the prevalence of HPV in extra-anatomical sites. For this reason, in this study, participants who engage in other sexual practices were not excluded.]

2) The quality of Figure 2 is unsatisfactory for analysis.

Response: [Thank you very much for your comment. The image was adjusted because the agarose gels obtained from the RFLP were unclear. We decided to include representative images about the integrity of DNA samples, internal control, viral DNA identification using the L1C1/L1C2 primers, and one representative image of the RFPL. However, if these figures are not considered relevant to the study, we can omit them.]

3) The figure legend lists types 33, 35, and 39. How does this relate to Table

Response: [Thank you for your comment. We apologize for the confusion. Numbers 33, 35, and 39 referred to the sample number in which HPV genotypes were identified, not to the genotypes themselves. This information was removed from the figure to avoid confusion and ensure clarity.]

4. Response to Comments on the Quality of English Language

Point 1: (x) The English is fine and does not require any improvement.

Response 1: (We appreciate this kind comment. Some details that were unclear in the manuscript have been improved to enhance the quality of the language in the text.)

[We greatly appreciate all the comments, observations, and suggestions made by the reviewers, which allowed us to conduct a thorough and in-depth review of the work. We hope that this is reflected in the observations included in this new version of the manuscript.]

Reviewer 2 Report

Comments and Suggestions for Authors

P1 abstract “a high percentage of oral and pharyngeal cancer cases in young individuals” might be misleading as both high percentage and young can be misinterpreted. Possibly the authors meant to highlight the observed raise in incidence in patients younger than 45?

P2 introduction, grammar issues “while HPV 13 is associated with oral hyperplasia focal epithelial,…”

P2 the sentence is unclear “Samples were randomly collected for one week following an "HPV screening campaign" conducted during the month prior to the study”. This is at odds with figure 1 which suggests that samples were collected over several months. It is unlikely that any randomization was done so the “randomly collected” might be inappropriate statement.

P3 small typos “MgCl2” 2 should be subscript, “extension step at 72ºC”  º symbol appears to have underline markup visible

P4 figure 1 shows some issues. Ethical approval should usually be obtained before approaching patients (so I suggest removing it from the figure), exclusion of acute respiratory infections should be done before sampling (ie to protect personnel collecting samples). It makes no sense to exclude those 4 patients after the samples were already collected since influenza is unlikely to affect HPV results.

P4 figure 1 caption has a typo “HPV by PCR-R.” should be RFLP

P4 results section. Consider rounding the numbers to 1 decimal point (except p values)

P4 “Of the HPV-positive cases, four (2.90%) reported genital warts” unclear what the percentages refer to. The results section often makes it challenging to understand what percentages are show but here the text is completely at odds as there were 6 positive cases.

P4 “(kissing with tongue), with a median of 5”  what does the median 5 refer to?

P4 the whole sentence  “About tobacco, alcohol, and drug use,…” should be checked for grammar and clarity

P4 section 3.1 should include a table summarizing the data either as a new table 1 or as a supplement table.

P4 Table 1 title omits HPV. Consider rounding numbers to 1 decimal point and p values to 3 points. “Significance” column is obsolete. Alcohol consumption has missing decimal point separator.

P5 section 3.2 title has a typo “HVP”. Order of sections is poor since 3.1 compares positive and negative samples while 3.2 establishes positivity itself.

P5 Figure 2 is better suited for a supplement. Figure 2 panels should all have names. There is no direct link between sample ID numbers shown on left panel vs sample IDs shown on subpanels A,b,c. No positive controls or referent RFLP patterns are shown for panels A,b,c making the figure less informative.

P5 Table 2 reference format in the table is different from that of the text. Furthermore it is unlikely that the association of HPV 11 with condylomas was first reported in 2024. Best practice would be to cite either the original contribution or the most recent comprehensive review on the topic.

P5 “Oral and oropharyngeal squamous cell papillomas (OPSCC)”  SCC is usually the abbreviation for squamous cell carcinoma. Possibly the authors meant OPSCP? However, this abbreviation is not used later so shouldn’t be introduced at all.

P5 discussion section. The first paragraph has grammar issues. First 2 sentences refer to lesions, while the 3rd sentence states “present study identified a prevalence of 4.4% in the oropharyngeal cavity” without clarifying that the topic under discussion went from lesions to HPV prevalence. This paragraph should be rephrased for clarity.

P5 “two genotypes considered  low / middle (11 and 13) were” sentence appears incomplete (ie lacks low/middle risk)

P6 the limitations paragraph is unclear at places and should be rewritten in full

Comments on the Quality of English Language

sentence structure is inappropriate or unclear at some places of the manuscript (see above)

Author Response

Comments 1: [P1 abstract “a high percentage of oral and pharyngeal cancer cases in young individuals” might be misleading as both high percentage and young can be misinterpreted. Possibly the authors meant to highlight the observed raise in incidence in patients younger than 45?.]

Response 1: [We agree with this comment. According to the reviewer, the text refers to the fact that the incidence of oral and pharyngeal cancer has increased in the population, which is due, among other factors, to new sexual practices, with young people being particularly affected. The introduction has been modified to avoid this confusion.]

Comments 2: [P2 introduction, grammar issues “while HPV 13 is associated with oral hyperplasia focal epithelial,…”.]

Response 2: [Thank you for pointing that out. The grammatical error has been corrected in the introduction.]

Comments 3: [P2 the sentence is unclear “Samples were randomly collected for one week following an "HPV screening campaign" conducted during the month prior to the study”. This is at odds with figure 1 which suggests that samples were collected over several months. It is unlikely that any randomization was done so the “randomly collected” might be inappropriate statement.]

Response 3: [Thank you very much for your suggestion. The information in the figure and methodology has been corrected, as the period indicated from August 2023 to January 2024 corresponds to the period during which the students were enrolled in the course, not the period during which the study was conducted. The study period was one week, following an awareness campaign. Similarly, the statement about random sampling was corrected, since all students who met the inclusion criteria were considered.]”

Comments 4: [P3 small typos “MgCl2” 2 should be subscript, “extension step at 72ºC”  º symbol appears to have underline markup visible]

Response 4: [Agree. Thank you for the observation, which has already been changed in the methodology.]”

Comments 5: [P4 figure 1 shows some issues. Ethical approval should usually be obtained before approaching patients (so I suggest removing it from the figure), exclusion of acute respiratory infections should be done before sampling (ie to protect personnel collecting samples). It makes no sense to exclude those 4 patients after the samples were already collected since influenza is unlikely to affect HPV results.]

Response 5: [Thank you very much for your comment. We would like to clarify that the record was obtained prior to the study, so the figure was modified to avoid confusion. With regard to the criterion of “excluding patients with respiratory infections,” the reason for this was based on previous studies, which reported that the presence of respiratory infections can cause contamination and introduce a viral load of other viruses or bacteria into the saliva and mucus sample, which interferes with molecular tests for HPV detection by PCR-RFLP, potentially causing false negatives.]”

Comments 6: [P4 figure 1 caption has a typo “HPV by PCR-R.” should be RFLP]

Response 6: [Agree. Thank you for the observation, which has already been modified in the figure caption.]”

Comments 7: [P4 results section. Consider rounding the numbers to 1 decimal point (except p values)]

Response 7: [Thank you very much for the suggestion. Table 3 was adjusted to one decimal point for the OR value and three decimal points for the p value, for greater clarity.]”

Comments 8: [P4 “Of the HPV-positive cases, four (2.90%) reported genital warts” unclear what the percentages refer to. The results section often makes it challenging to understand what percentages are show but here the text is completely at odds as there were 6 positive cases.]

Response 8: [Thank you for your comment. This point refers to those participants who reported a previous HPV infection in the survey. Of the total population (136 participants), five reported a previous HPV infection, of which four indicated genital warts, corresponding to 2.9% of the total population. The information has been modified for clarity.]”

Comments 9: [P4 “(kissing with tongue), with a median of 5”  what does the median 5 refer to?]

Response 9: [Thank you for pointing that out. We apologize for the confusion. The term “median” refers to the average number of individuals with whom participants reported having “kissing with tongue” throughout their sexual lives. The paragraph has been modified to avoid confusion. ]”

Comments 10: [P4 the whole sentence  “About tobacco, alcohol, and drug use,…” should be checked for grammar and clarity]

Response 10: [Thank you very much for your comment. The grammar in the paragraph has been revised and modified to clarify the sentence.]”

Comments 11: [P4 section 3.1 should include a table summarizing the data either as a new table 1 or as a supplement table.]

Response 11: [Thank you for the suggestion. A table (Table 2) has been included, summarizing the general characteristics of the population.]”

Comments 12: [P4 Table 1 title omits HPV. Consider rounding numbers to 1 decimal point and p values to 3 points. “Significance” column is obsolete. Alcohol consumption has missing decimal point separator.

Response 12: [Thank you very much for your feedback. The comments suggested have been addressed in the table. In this new version, it is labeled as Table 3. The “significance” was omitted.]”

Comments 13: [P5 section 3.2 title has a typo “HVP”. Order of sections is poor since 3.1 compares positive and negative samples while 3.2 establishes positivity itself.]

Response 13: [Thank you for your comment. We apologize for the confusion. We would like to clarify that Section 3.1 describes the clinical characteristics of the population reported by participants through a questionnaire applied before saliva samples were taken. Section 3.2 corresponds to the molecular results obtained in this study. ]”

Comments 14: [P5 Figure 2 is better suited for a supplement. Figure 2 panels should all have names. There is no direct link between sample ID numbers shown on left panel vs sample IDs shown on subpanels A,b,c. No positive controls or referent RFLP patterns are shown for panels A,b,c making the figure less informative..]

Response 14: [Thank you very much for the suggestion. The figures were replaced due to their lack of clarity, and only representative images were included to verify the expected results. However, if necessary, the images can be omitted or included as supplementary material.]”

Comments 15: [P5 Table 2 reference format in the table is different from that of the text. Furthermore it is unlikely that the association of HPV 11 with condylomas was first reported in 2024. Best practice would be to cite either the original contribution or the most recent comprehensive review on the topic..]

Response 15: [Thank you very much for your comment. The reference format has been modified. We would like to clarify that the references included in this table correspond to reports that are consistent with observations made in young populations, both globally and in Mexico. The aim of this was to make an accurate comparison of the results observed in young populations, particularly in Mexico.]”

Comments 16: [P5 “Oral and oropharyngeal squamous cell papillomas (OPSCC)”  SCC is usually the abbreviation for squamous cell carcinoma. Possibly the authors meant OPSCP? However, this abbreviation is not used later so shouldn’t be introduced at all.]

Response 16: [Thank you very much for pointing that out. The abbreviation has been removed to avoid confusion.]”

Comments 17: [P5 discussion section. The first paragraph has grammar issues. First 2 sentences refer to lesions, while the 3rd sentence states “present study identified a prevalence of 4.4% in the oropharyngeal cavity” without clarifying that the topic under discussion went from lesions to HPV prevalence. This paragraph should be rephrased for clarity..]

Response 17: [Thank you for pointing that out. We apologize for the confusion. We have rephrased this section of the discussion to clarify the information.]”

Comments 18: [P5 “two genotypes considered  low / middle (11 and 13) were” sentence appears incomplete (ie lacks low/middle risk).]

Response 18: [Thank you for your comment. The information was reviewed to complete the statement and ensure that it does not appear incomplete.]”

Comments 19: [P6 the limitations paragraph is unclear at places and should be rewritten in full]

Response 19: [Thank you for your comment. The limitations section was reviewed and rewritten to avoid confusing information.]”

4. Response to Comments on the Quality of English Language

Point 1: (x) The English could be improved to more clearly express the research

Response 1: (We appreciate this comment. The manuscript has been revised to improve the quality of the language in the text.)

5. Additional clarifications

[We greatly appreciate all the comments, observations, and suggestions made by the reviewers, which allowed us to conduct a thorough and in-depth review of the work. We hope that this is reflected in the observations included in this new version of the manuscript.]

Round 2

Reviewer 1 Report

Comments and Suggestions for Authors

The authors of the manuscript have done a significant job of addressing comments. In its current form, the study is of interest to the entire medical community and significantly expands our understanding of the characteristics of HPV circulation in the oropharynx in a southwest young population in Mexico.
The revised article is recommended for publication.